# When Less is More: Simplifying Inputs Aids Neural Network Understanding

## Abstract

How do neural network image classifiers respond to simpler and simpler inputs? And what do such responses reveal about the characteristics of the data and their interaction with the learning process? To answer these questions, we need measures of input simplicity (or inversely, complexity) and class-discriminative input information as well as a framework to optimize these during training. Lastly, we need experiments that evaluate whether this framework can simplify data to remove injected distractors and evaluate the impact of such simplification on real-world data. In this work, we measure simplicity with the encoding bit size given by a pretrained generative model, and minimize the bit size to simplify inputs during training. At the same time, we minimize a gradient- and activation-based distance metric between original and simplified inputs to retain discriminative information. We investigate the trade-off between input simplicity and task performance. For images with injected distractors, such simplification naturally removes superfluous information. For real-world datasets, qualitative analysis suggests the simplified images retain visually discriminative features and quantitative analysis show they retain features that may be more robust in some settings.

## 1 Introduction

A better understanding of the information deep neural networks use to learn can lead to new scientific discoveries (Raghu & Schmidt, 2020), highlight differences between human and model behaviors (Makino et al., 2022) and serve as powerful auditing tools (Geirhos et al., 2020; D'souza et al., 2021; Bastings et al., 2021; Agarwal et al., 2021).

Removing information from the input deliberately is one way to illuminate what information content is relevant for learning. For example, occluding specific regions, or removing certain frequency ranges from the input gives insight into which input regions and frequency ranges are relevant for the network's prediction (Zintgraf et al., 2017; Makino et al., 2022; Banerjee et al., 2021a). These ablation techniques use simple heuristics such as random removal (Hooker et al., 2019; Madsen et al., 2021), or exploit domain knowledge about interpretable aspects of the input to create simpler versions of the input on which the network's prediction is analyzed (Banerjee et al., 2021b).

What if, instead of using heuristics, one *learns* to synthesize simpler inputs that retain task-relevant information? This way, we could gain intuition into the model behavior without relying on prior or domain knowledge about what input content may be relevant for the network's learning target. To achieve this, one needs to define the precise meaning of "simplifying an input" and "retaining task-relevant information", including metrics for simplicity and retention of task-relevant information.

In this work, we propose *SimpleBits*, an information-reduction method that learns to synthesize simplified inputs which contain less information while remaining informative for the task. To measure simplicity, we use a finding initially reported as a problem for density-based anomaly detection—that generative image models tend to assign higher probability densities and hence lower bits to visually simpler inputs (Kirichenko et al., 2020; Schirrmeister et al., 2020). Here, we use this to our advantage and minimize the encoding bit size given by a generative network trained on a general image distribution to simplify inputs. To measure the

retention of task-relevant information, we analyze several activation- and gradient-based metrics for input similarity. Our analysis shows that gradient-weighted activation differences between the network activations of the simplified and the original input are a suitable metric.

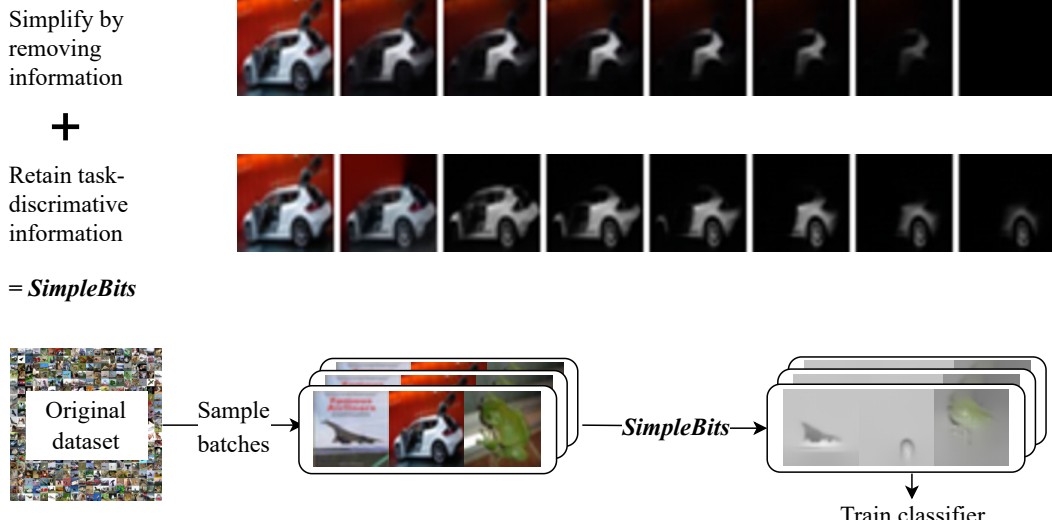

Figure 1: *SimpleBits* overview. Top: *SimpleBits* simplifies images by removing information as measured by a generative model while also retaining task-discriminative information, see Section 2 and 3. Wheel of the car remains visible after *SimpleBits* simplification. Bottom: We apply *SimpleBits* as a per-instance simplifier during training, where each image is simplified but the total number of images remain the same.

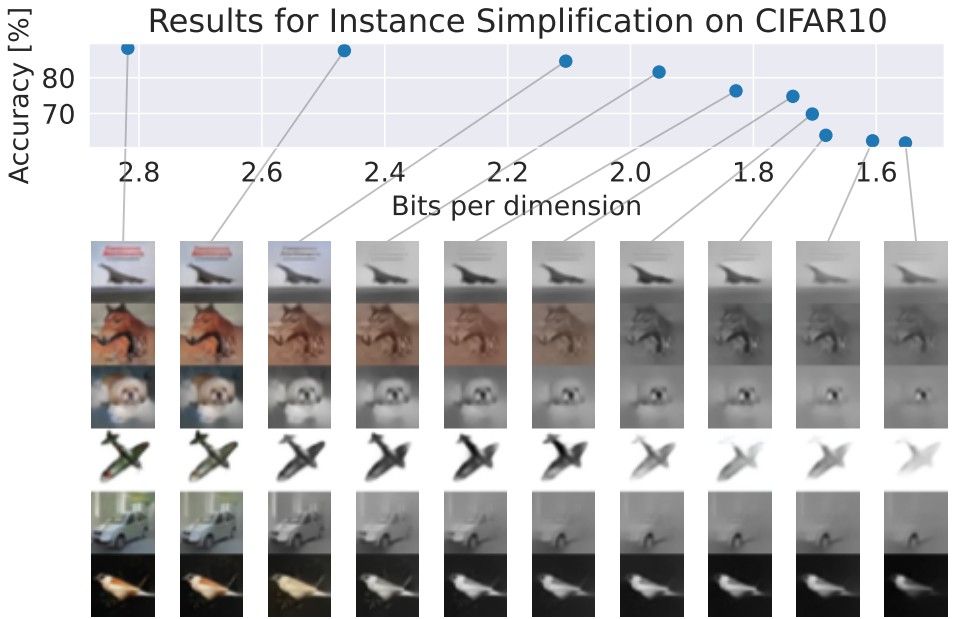

Figure 2: Examples of *SimpleBits* applied to CIFAR10 training images. Simplified images with varying degree of simplication by varying upper bound on $L_{\text{task}}$, see Section 4. We observe that at lower bits per dimension, only few discriminative features remain in the image.

We apply *SimpleBits* in a *per-instance* setting, where each image is processed to be a simplified version of itself during training, see Figure 1. We use a variety of synthetic and real-world image datasets to evaluate

the effect of *SimpleBits* on the network behavior and on the simplified input data. Our synthetic datasets contain injected noise distractors that *SimpleBits* is supposed to remove, allowing a controlled evaluation of the *SimpleBits* framework. For real-world datasets, we analyze the tradeoff between simplification and task performance and the characteristics of the resulting simplified images, see Figure 2 for CIFAR10 examples.

Our evaluation provides the following insights:

1. **Successful distractor removal.** *SimpleBits* successfully removes superfluous information for tasks with injected distractors.

2. **Visually task-specific simplified images.** On natural image datasets, qualitative analysis suggests *SimpleBits* retains plausible task-relevant information across all datasets we tried.

3. **Dataset-dependent simplification effects on accuracy.** Increasing simplification leads to accuracy decreases, as expected, and we report the trade-off between input simplification and task performance for different datasets and problems.

4. **SimpleBits learns to represent robust features in the simplified images.** Evaluations on CIFAR10-C, which consists of the CIFAR10 evaluation set with various corruptions applied (Hendrycks & Dietterich, 2019), suggest that *SimpleBits* retains robust features.

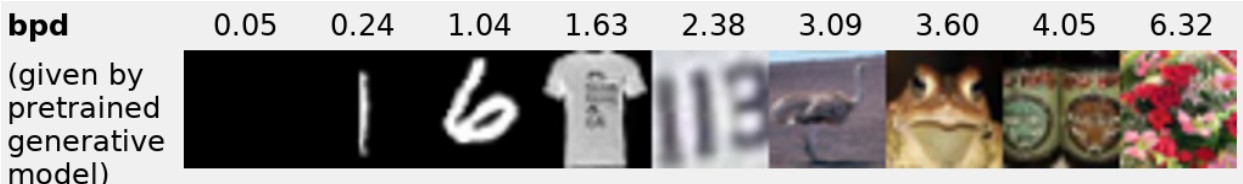

Figure 3: Visualization of the bits-per-dimension (bpd) measure for image complexity, sorted from low to high. Image samples are taken from MNIST, Fashion-MNIST, CIFAR10 and CIFAR100, in addition to a completely black image sample. bpd is calculated from the density produced by a Glow (Kingma & Dhariwal, 2018) model pretrained on 80 Million Tiny Images. Switching to other types of generative models including PixelCNN and diffusion models trained on other datasets produces consistent observations; see Figure S1 in Supplementary Information for more details.

## 2 Measuring and Reducing Instance Complexity

How to define simplicity? We use the fact that generative image models tend to assign lower encoding bit sizes to visually simpler inputs (Kirichenko et al., 2020; Schirrmeister et al., 2020). Concretely, the complexity of an image $x$ can be quantified as the negative log probability mass given by a pretrained generative model with tractable likelihood, $G$, i.e. $-\log p_G(x)$. $-\log p_G(x)$ can be interpreted as the image encoding size in bits per dimension (bpd) via Shannon's theorem (Shannon, 1948): $\text{bpd}(x) = -\log_2 p_G(x)/d$ where $d$ is the dimension of the flattened $x$.

The simplification loss for an input $x$, given a pre-trained generative model $G$, is as follows:

$$L_{\text{sim}}(x) = -\log p_G(x) \tag{1}$$

Figure 3 visualizes images and their corresponding bits-per-dimension (bpd) values given by a Glow network (Kingma & Dhariwal, 2018) trained on 80 Million Tiny Images (Torralba et al., 2008) (see Supplementary Section S1 for other models). This is the generative network used across all our experiments. A visual inspection of Figure 3 suggests that lower bpd corresponds with simpler inputs, as also noted in prior work (Serrà et al., 2020). The goal of our approach, *SimpleBits*, is to minimize bpd of input images while preserving task-relevant information. In the following section, we will explore how to ensure the preservation of task-relevant information.

# 3   Measuring And Preserving Task-Relevant Instance Information

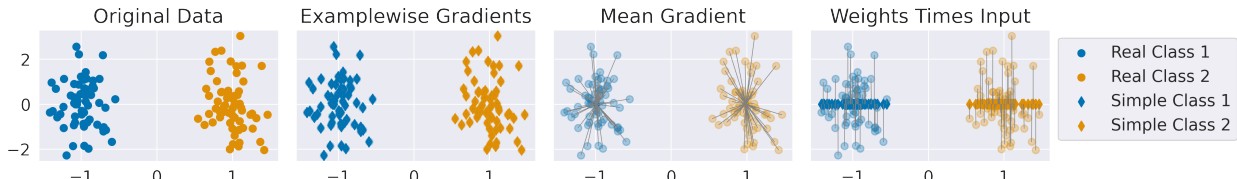

Figure 4: Result of matching different values on 2D binary classification problem with one discriminative and one non-discriminative dimension. Matching the examplewise gradients leads to retaining nondiscriminative information, matching the average gradient leads to losing the correspondence between individual simplified-original example pairs, whereas matching weights times input leads to the desired behavior.

How to measure the preservation of task-relevant information? The task-relevant information learned by a network must be reflected in the network activations and hence minimizing the effect of the simplification on them may be a viable approach. However, different activations vary in their contribution to the final prediction, due to the later processing in subsequent layers. Therefore, we will weight activation changes by the gradient of the classification loss function in our information-preservation measure.

We propose to use the distance of the gradient-weighted network activations of the original and the simplified input as our information-preservation measure. The gradient of the loss on the original input with regard to the network activations $\frac{\partial L(f(x_{orig}),y))}{\partial h(x_{orig})}$ is used as a proxy for the contribution of that activation to the prediction $f(x_{orig})$. Therefore, we try to reduce the distance of the activations weighted by that gradient, i.e., reducing the distances of $v_{orig} = \frac{\partial L(f(x_{orig}),y))}{\partial h(x_{orig})} \circ h(x_{orig})$ and $v_{simple} = \frac{\partial L(f(x_{orig}),y))}{\partial h(x_{orig})} \circ h(x_{simple})$.

A variety of distance functions have recently been used to match gradients or weight changes, e.g., layerwise cosine similarity (Zhao et al., 2021) or l2-normalized squared distance (Cazenavette et al., 2022). Here, we use layerwise l2-normalized squared distance, concretely $\frac{||v^l_{simple} - v^l_{orig}||_2^2}{||v^l_{orig}||_2^2}$ for original and simplified values for layer $l$, so for $v^l_{orig} = \frac{\partial L(f(x_{orig}),y))}{\partial h^l(x_{orig})} \circ h^l(x_{orig})$ and $v^l_{simple} = \frac{\partial L(f(x_{orig}),y))}{\partial h^l(x_{orig})} \circ h^l(x_{simple})$. Using the layerwise l2-normalized squared distance instead of the cosine distance takes into account the magnitude of the activations, which preliminary experiments showed to better ensure that the simplified images remain visually interpretable. Overall, this results in the loss function:

$$L_{\text{task}}(x_{simple}) = \sum_l \frac{|| \frac{\partial L(f(x_{orig}),y))}{\partial h^l(x_{orig})} \circ (h^l(x_{orig}) - h^l(x_{simple}))||_2^2}{|| \circ \frac{\partial L(f(x_{orig}),y))}{\partial h^l(x_{orig})} \circ h^l(x_{orig})||_2^2} \tag{2}$$

We will further motivate our measure with a 2D toy example. Our 2D binary classification problem has one discriminative dimension (both classes have different distributions) and one nondiscriminative dimension (both classes have the same distribution), see Fig. 4. Here, we propose the simplification should only retain the input values of the discriminative dimension and map the nondiscriminative dimension to zero.

Commonly used methods to preserve information by matching gradients (as used in dataset condensation methods that condense entire training datasets into smaller synthetic datasets) will fail to achieve the desired result. Matching the per-example loss gradients undesirably retains the values of the nondiscriminative dimension in the simplified examples. The loss gradient for example $x$ and target $y$ wrt. to weights $w$ is $\frac{\partial L(f(x),y)}{\partial w} = \frac{\partial L(f(x),y)}{\partial f(x)} \cdot \frac{\partial f(x)}{\partial w} = \frac{\partial L(f(x),y)}{\partial f(x)} \cdot x$, so both dimensions $x_1$ and $x_2$ will be optimized to be the same for the original and simplified examples. Instead matching the gradient averaged over all examples maps all simplified examples of one class to the same point, losing the correspondence between simplified and original examples. What does work as desired is matching $w \circ x$ as seen in Fig. 4, where only the

---

**Algorithm 1** One SimpleBits Joint Training Step

---
1: **given** generative network $G$, input and target batch $\boldsymbol{X}$ and $\boldsymbol{y}$, the classifier clf and the image-to-image simplifier network.
2: *Create a copy of the classifier and train it one step on real data, e.g. using SGD/Adam*
3: copied_clf = train_one_step(copy(clf), $\boldsymbol{X}$, $\boldsymbol{y}$)
4: *Simplify original data batch*
5: $\boldsymbol{X}_{\text{simple}}$ = simplifier($\boldsymbol{X}$)
6: *Compute per-example simplification loss as in equation 1*
7: **sim_loss** = $\log p_G(\boldsymbol{X}_{\text{simple}})$
8: *Compute per-example task loss as in equation 2*
9: **task_loss** = $L_{\text{task}}$(copied_clf, $\boldsymbol{X}_{\text{simple}}$, $\boldsymbol{X}$, $\boldsymbol{y}$)
10: *Predict original data to know which examples are predicted correctly.*
11: **predicted_y** = copied_clf($\boldsymbol{X}$)
12: *Mask out simplification loss for examples with too high task loss or incorrectly predicted class.*
13: **sim_loss** = **sim_loss** $\circ$ (**task_loss** < threshold) $\circ$ (**predicted_y** == $\boldsymbol{y}$)
14: *Compute average total loss over examples.*
15: total_loss = average(**sim_loss** + **task_loss**)
16: *Update simplifier using the total loss, e.g. with SGD/Adam*
17: simplifier = update_simplifier(simplifier, total_loss)
18: *Simplify original data batch with updated simplifier*
19: $\boldsymbol{X}_{\text{simple}}$ = simplifier($\boldsymbol{X}$)
20: *Train one step with classification loss on simplified data, e.g. using SGD/Adam.*
21: clf = train_one_step(clf, $\boldsymbol{X}_{\text{simple}}$, $\boldsymbol{y}$)
22: *Return updated simplifier and classifier*
23: **return** simplifier, clf

---

discriminative dimension $x_1$ is optimized to be the same for the simplified and the original examples. Noting that the gradient of loss of the with regard to the inputs is $\frac{\partial L(f(x),y)}{\partial x} = \frac{\partial L(f(x),y)}{\partial f(x)} \circ \frac{\partial f(x)}{\partial x} = \frac{\partial L(f(x),y)}{\partial f(x)} \circ w$, our gradient-weighed activation matching can be seen as a generalization of the $x \circ w$ matching to deep networks.

In preliminary work, we also considered $w \circ \frac{\partial L(f(x),y)}{\partial w}$ (weights times gradients wrt. to weights). We mainly opted for the activation-based matching due to the faster computation as no examplewise weight gradients need to be computed. Note that for the simple linear model derived above, both of these metrics are equivalent $\frac{\partial L(f(x),y)}{\partial w} \circ w = \frac{\partial L(f(x),y)}{\partial f(x)} \cdot \frac{\partial f(x)}{\partial w} \circ w = \frac{\partial L(f(x),y)}{\partial f(x)} \cdot x \circ w = \frac{\partial L(f(x),y)}{\partial f(x)} \cdot x \circ \frac{\partial f(x)}{\partial x} = \frac{\partial L(f(x),y)}{\partial x} \cdot x$. We note that both of these metrics have been used elsewhere, e.g. the $w \circ \frac{\partial L(f(x),y)}{\partial w}$ metric has been used to identify important connections for pruning (Lee et al., 2019) and $h(x) \circ \frac{\partial L(f(x),y))}{\partial h(x)}$ has been used in interpretability work where mostly the original input activation $x$ was used as $h(x)$ (Shrikumar et al., 2017).

## 4    Per-Instance Simplification During Training

When plugged into the training of a classifier $f$, *SimpleBits* aims to simplify each image such that $f$ can still learn the original classification task from the simplified images. For that, we train an image-to-image network simplifier to simultaneously optimize $L_{\text{sim}}$ and $L_{\text{task}}$ while we train the classifier on the simplified images. As the classifier guides the simplifier in learning what information is task-relevant, we also need to enable the classifier to learn task-relevant information that the simplifier network has not included in the simplified images at that point. For that, before computing $L_{\text{task}}$ on a paired batch of real and simplified images, we first copy the current classifier state and train that copied classifier for one step on the real images of that batch. Since only a copy of the classifier is trained in that step, the actual classifier still only learns from the simplified images.

To achieve a tradeoff between simplification and preservation of task-relevant information, we set an upper bound on the allowed $L_{\text{task}}$. This intuitively defines how much task-relevant information may be lost due to the simplification on each example and we vary it to investigate different tradeoffs. We only optimize the simplification loss on an example as long as (a) the $L_{\text{task}}$ stays below the threshold on that same example (b) the classifier predicts the correct class on that example. This was empirically more stable than using a linear combination of the losses and also had the desired result of keeping task-relevant information in each image rather than just in the dataset. Pseudocode for one training step can be found in Algorithm 1.

# 5 Experiments

Our classifier architecture is a normalizer-free architecture to avoid interpretation difficulties that may arise from normalization layers, such as image values being downscaled by the simplifier and then renormalized again. We use Wide Residual Networks as described by Brock et al. (2021). The normalizer-free architecture reaches 94.0% on CIFAR10 in our experiments, however we opt for a smaller variant for faster experiments that reaches 91.7%; additional details are included in the Supplementary Section S2.1. For the simplifier network, we extend the Wide ResNet architecture with 3 output heads at different stages of the network that outputs encodings to be inverted by our generative model, a three-scale Glow architecture as mentioned in Section 2. This leverages the learnt decoder of the Glow architecture for our simplifier, more details are included in Supplementary Section S2.2.

## 5.1 *SimpleBits* Removes Injected Distractors

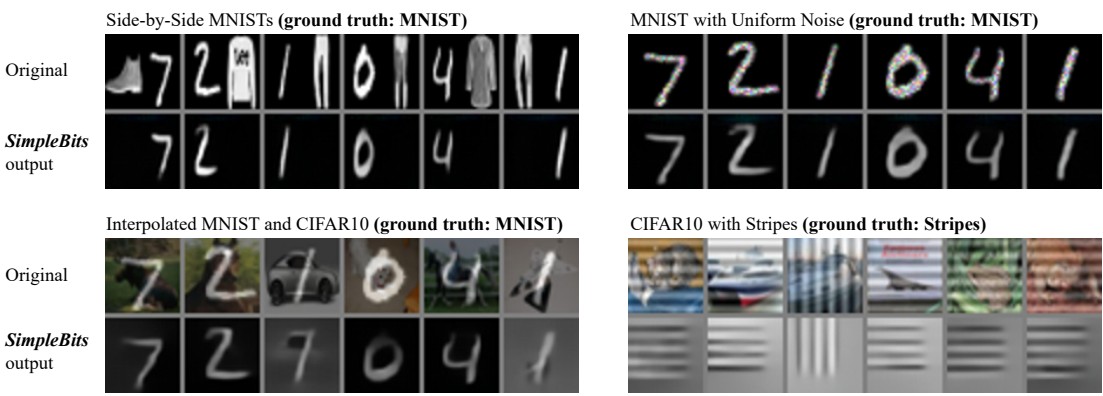

Figure 5: Evaluation of *SimpleBits* as a distractor removal on four composite datasets. Shown are the composite original images and the corresponding simplified images produced by *SimpleBits* trained alongside the classifier. *SimpleBits* is able to almost entirely remove task-irrelevant image parts, namely FashionMNIST (**top left**), random noise (**top right**), CIFAR10 (**bottom left** as well as **bottom right**).

We first evaluate whether our per-instance simplification during training successfully removes superfluous information for tasks with injected distractors. To that end, we construct datasets to contain both useful (ground truth) and redundant (distractor) information for task learning. We create four composite datasets derived from three conventional datasets: MNIST (LeCun & Cortes, 2010), FashionMNIST (Xiao et al., 2017) and CIFAR10 (Krizhevsky, 2009). Sample images, both constructed (input to the whole model) and simplified (output of simplifier and input to classifier), are shown in Figure 5.

**Side-by-Side MNIST** constructs each image by horizontally concatenating one image from Fashion-MNIST and one from MNIST. Each image is rescaled to 16x32, so the concatenated image size remains 32x32; the order of concatenation is random. The ground truth target is MNIST labels, and therefore FashionMNIST is an irrelevant distractor for the classification task. As seen in Figure 5, the simplifier effectively removes the clothes side of the image.

**MNIST with Uniform Noise** adds uniform noise to the MNIST digits, preserving the MNIST digit as the classification target. Hence the noise is the distractor and is expected to be removed. And indeed the noise is no longer visible in the simplified outputs shown in Figure 5.

**Interpolated MNIST and CIFAR10** is constructed by interpolating between MNIST and CIFAR10 images. MNIST digits are the classification target. The expectation is that the simplified images should no longer contain any of the CIFAR10 image information. The result shows that most of the CIFAR10 information is removed, leaving only some information that is visually similar to parts of digits.

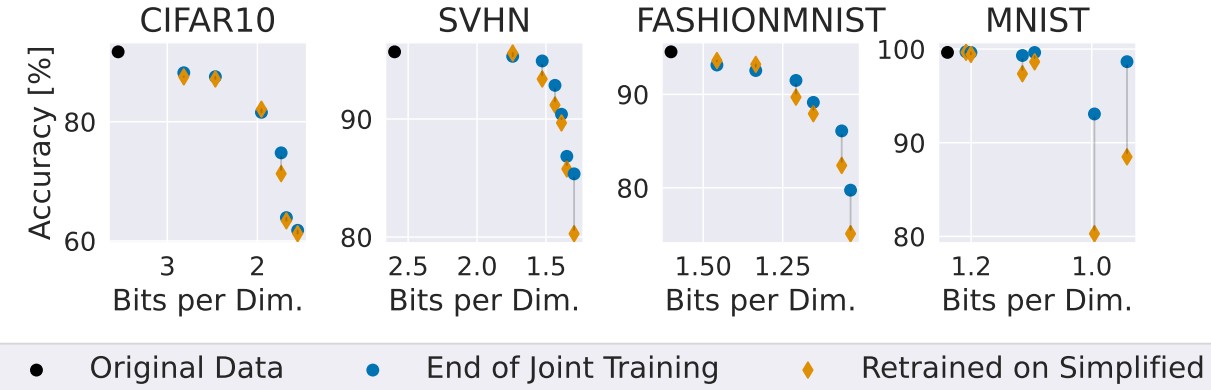

Figure 6: Results for training image simplifications on real datasets. Dots show results for training with varying upper bound on $L_{\text{task}}$. Images with less bits per dimension lead to reduced accuracies, and such reduction is more pronounced for complex datasets like CIFAR10.

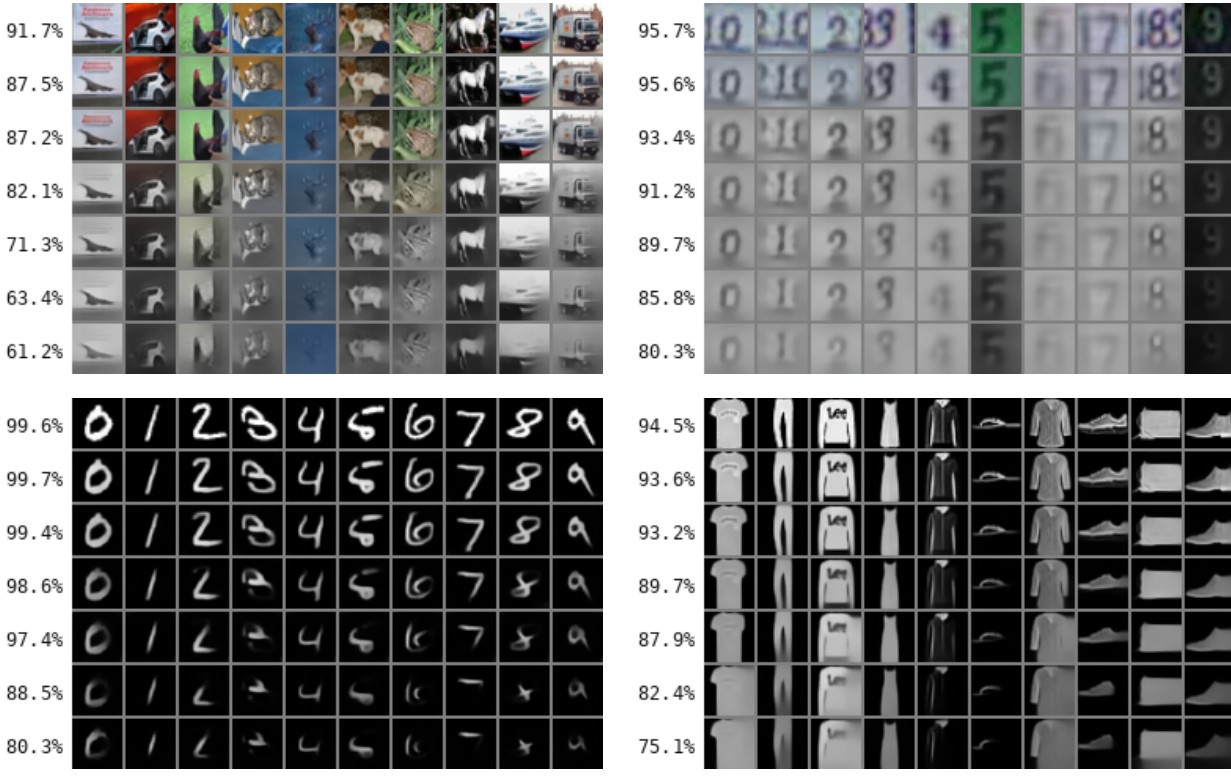

Figure 7: *SimpleBits* examples of training for increased simplification by increasing loss bound for $L_{task}$ on CIFAR10, SVHN, MNIST and Fashion-MNIST. Images produced by the simplifier at end of joint training, shown are the first example of each class in the test set. Percentages left of each row indicate accuracies after retraining on images produced by the same simplifier.

**CIFAR10 with Stripes** overlays either horizontal or vertical stripes onto CIFAR10 images, with the binary classification label 0 for horizontal and 1 for vertical stripes. With this dataset the simplification mostly only retains some horizontal and vertical stripes, which is enough to solve this task.

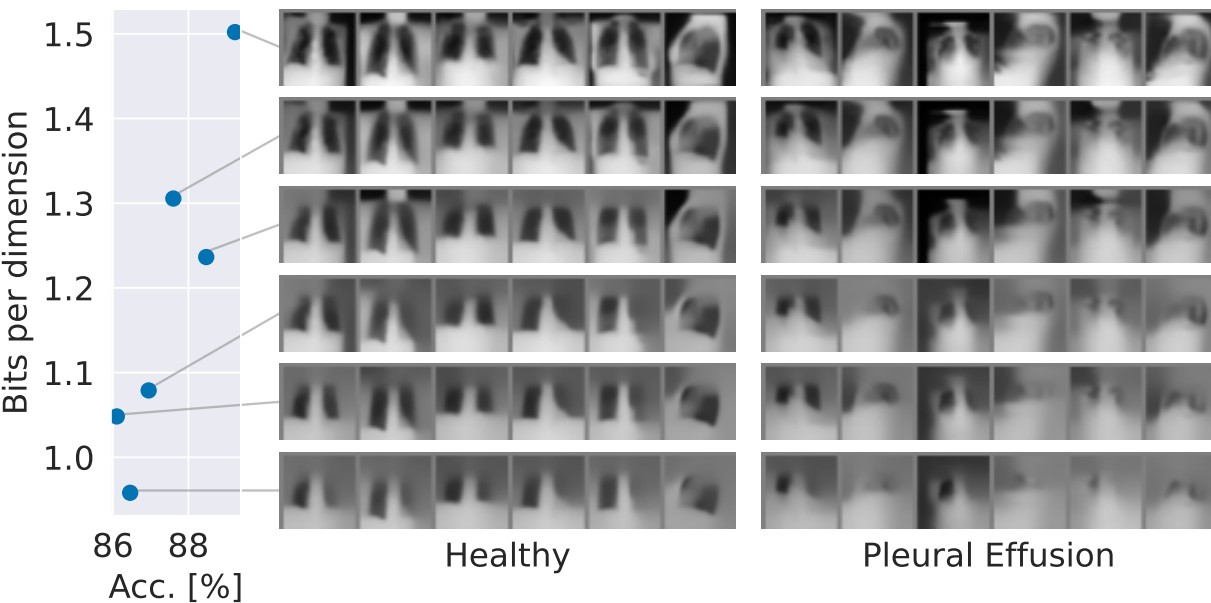

Figure 8: Results on MIMIC-CXR Pleural Effusion vs. Healthy Diagnosis from Chest X-Rays.

## 5.2 Trade-off Curves on Conventional Datasets

We verified that *SimpleBits* is able to discern information relevance in inputs, and effectively removes redundant content to better serve the classification task. In real-world datasets, however, the type, as well as the amount of information redundancy in inputs is often unclear.

With the same framework, by varying the upper bound on the allowed $L_{\text{task}}$ we can study the trade-off between task performance and level of simplification to better understand a given dataset. With lower upper bounds, the training should resemble conventional training as $L_{\text{sim}}$ will be mostly turned off. On the other end, with higher upper bounds, the inputs should be more drastically simplified at the expense of accuracy due to the loss of task-relevant information.

We experimented with MNIST, Fashion-MNIST, SVHN (Netzer et al., 2011) and CIFAR10, producing a trade-off curve for each by setting the upper bound on the allowed $L_{\text{task}}$ various values for different training runs. For each setting, we report the classification accuracy as a result of joint training of simplification and classification and after retraining from scratch with only simplified images, as shown in Figure 6. We also show the first simplified test image of each class, see Figure 7.

As expected, higher strengths of simplification lead to decreased task performance. Interestingly, such decay is observed to be more pronounced for more complex datasets such as CIFAR10. This suggests either the presence of a relatively small amount of information redundancy, or that naturally occurring noise in data help with generalization, analogous to how injected noise from data augmentation helps learning features invariant to translations and viewpoints, even though the augmentation itself does not contain additional discriminative information.

Qualitative analysis of the simplified images in Figure 7 suggests that even with strong simplification, discriminative information is retained in the images. At the strongest simplification level, with many details removed, most classes are still visually identifiable.

We run three baselines to validate the efficacy of our simplification framework, as described in Supplementary Section S5 and shown in Figure S4. Section S3 shows that our simplification has practical savings in image storage. Training curves can be seen in Section S4. Additionally, we show that *SimpleBits* can also simplify images in a dataset condensation setting, see Section S8 .

## 5.3 *SimpleBits* is Able to Retain Good Accuracies on Corrupted CIFAR10

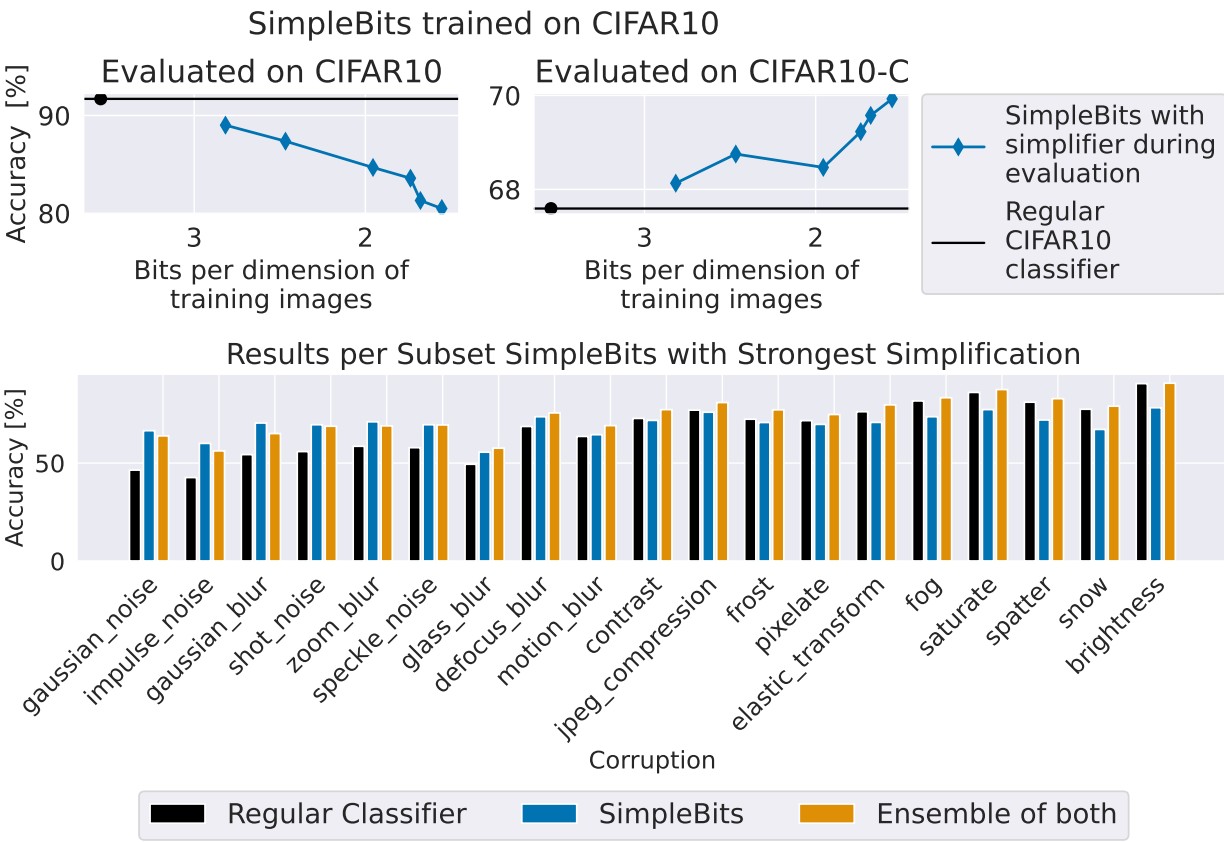

Figure 9: Results of *SimpleBits* on CIFAR10-C. **Top:** Accuracies of classifiers trained with varying upper bound on $L_{task}$ at the end of joint training. Blue lines indicate accuracies after *SimpleBits* training, with the frozen simplifier network applied to the test data during evaluation. Black lines indicate baseline accuracy of a network trained without any simplification. Accuracies decrease with increasing simplification on the CIFAR10 dataset, but accuracies do not decrease and even slightly increase for the simplified images on CIFAR10-C. This may indicate the combination of the simplifier and the classifier have learned more robust features. **Bottom:** Results per CIFAR10-C subset of the *SimpleBits*-trained model with strongest simplification. *SimpleBits* has especially high accuracies on high-frequency noise and blurring corruptions.

We evaluated the performance of our *SimpleBits*-trained CIFAR10 models on the CIFAR10-C dataset, which consists of the CIFAR10 evaluation set with various corruptions applied (Hendrycks & Dietterich, 2019). We found that *SimpleBits* evaluated on the simplified images actually increased accuracy on CIFAR10-C with increasing simplification, even slightly outperforming a regularly trained CIFAR10 classifier (see Figure 9). *SimpleBits* has especially high accuracies on high-frequency noise and blurring corruptions. This suggests that the combination of the simplifier and *SimpleBits*-trained classifier is able to retain features robust to some corruptions on CIFAR10.

## 5.4 *SimpleBits* Retains Plausible Features on Chest X-Rays

We applied *SimpleBits* to the MIMIC-CXR-JPG dataset to classify Chest X-Rays of healthy patients and those with pleural effusion. The classifier was able to maintain high accuracy across all levels of simplification, and the simplified images still displayed plausible Chest-X-Ray features for pleural effusion, see Figure 8.

# 6 Related Work

Our approach simplifying individual training images builds on Raghu et al. (2021), where they learn to inject information into the classifier training. Per-instance simplification during training can be seen as a instance of their framework combined with the idea of input simplification. In difference to their methods, *SimpleBits* explicitly aims for interpretability through input simplification.

Other interpretability approaches that synthesize inputs include generating counterfactual inputs (Hvilshøj et al., 2021; Dombrowski et al., 2021; Goyal et al., 2019) or inputs with exaggerated features (Singla et al., 2020). *SimpleBits* differs in explicitly optimizing the inputs to be simpler.

Generative models have often been used in various ways for interpretability such as generating realistic-looking inputs (Montavon et al., 2018) and by directly training generative classifiers (Hvilshøj et al., 2021; Dombrowski et al., 2021), but we are not aware of any work except (Dubois et al., 2021) (discussed above) to explicitly generate simpler inputs. A different approach to reduce input bits while retaining classification performance is to train a compressor that only keeps information that is invariant to predefined label-preserving augmentations. Dubois et al. (2021) implement this elegant approach in two ways. In their first variant, by training a VAE to reconstruct an unaugmented input from augmented (e.g. rotated, translated, sheared) versions. In their second variant, building on the CLIP (Radford et al., 2021) model, they view images with the same associated text captions as augmented versions of each other. This allows the use of compressed CLIP encodings for classification and achieves up to 1000x compression on Imagenet without decreasing classification accuracy. Their approach focuses on achieving maximum compression while our approach is focused on interpretability. Their approach requires access to predefined label-preserving augmentations and has reduced classification performance in input space compared to latent/encoding space.

# 7 Conclusion

We propose *SimpleBits*, an information-based method to synthesize simplified inputs. Crucially, *SimpleBits* does not require any domain-specific knowledge to constrain or dictate which input components should be removed; instead *SimpleBits* itself learns to remove the components of inputs which are least relevant for a given task.

Our simplification approach sheds light on the information required for a deep network classifier to learn its task. It can remove injected distractors and reveal discriminative features. Further, we find that for our approach the tradeoff between task performance and input simplification varies by dataset and setting — it is more pronounced for more complex datasets.

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

# Supplementary Information for:
# When Less is More: Simplifying Inputs Aids Neural Network Understanding

## Supplementary Outline

This document completes the presentation of the main paper with the following:

| Supplementary Section | Type of Content | Relevant Section in Main Text | TL;DR |
|---|---|---|---|
| S1 | *Additional experiments* | Section 2 | Example images with bpd values produced by other generative models |
| S2 | *Implementation details* | Section 4 | Architecture details for *simplification during training* |
| S3, S4, S5 | *Additional experiments* | Section 5.2 | Baselines, training curves, file size analysis |
| S7 | *Additional experiments* | Section 5.2 | *SimpleBits* applied to Vision Transformers |
| S6 | *More figures* | Figure 2 | Uncurated sets of simplified images at end of joint training |
| S8 | *Additional experiments* | - | *SimpleBits* applied to Dataset Condensation |

## S1    BPDs of Other Generative Models

Figure S1 shows that the bits per dimension produced by other generative models than Glow also correlate well with visual complexity, validating our measure. This is consistent with prior work that found bpds of generative models trained on natural image datasets are strongly influenced by general natural image characteristics independent of any specific dataset (Kirichenko et al., 2020; Schirrmeister et al., 2020; Havtorn et al., 2021).

## S2    Architecture Details for Per-Instance Simplification During Training

### S2.1    Classifier Network

Our classification network is based on the Wide ResNet architecture (Zagoruyko & Komodakis, 2016). We used a version with relatively few parameters with depth = 16 and widen_factor = 2 to allow for fast iteration on experimentation. We used ELU instead of ReLU nonlinearities.

Additionally, we removed batch normalization to avoid interference of normalization layers with the simplification process. We followed the method from Brock et al. (2021) to create a normalizer-free Wide ResNet. We reparameterize the convolutional layers using Scaled Weight Standardization:

$$\hat{W}_{ij} = \frac{W_{ij} - \mu_i}{\sqrt{N}\sigma_i}, \tag{S1}$$

where $\mu_i = (1/N)\sum_j W_{ij}$, $\sigma_i^2 = (1/N)\sum_j (W_{ij} - \mu_i)^2$, and $N$ denotes the fan-in. Further as in Brock et al. (2021), "activation functions are also scaled by a non-linearity specific scalar gain $\gamma$, which ensures

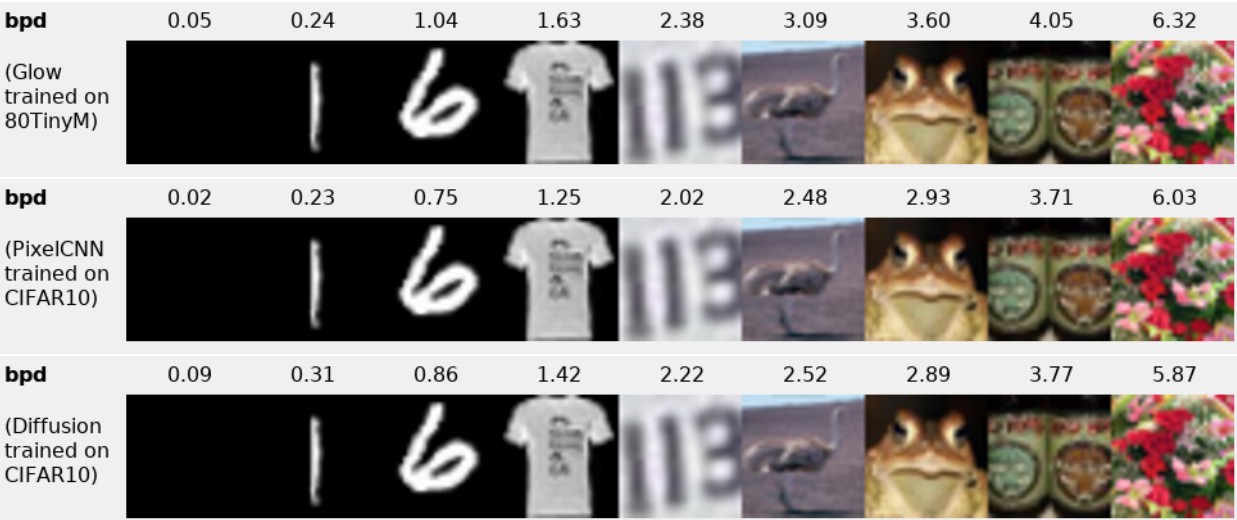

Figure S1: Visualization of the bits-per-dimension (bpd) measure for image complexity, sorted from low to high. Image samples are taken from MNIST, Fashion-MNIST, CIFAR10 and CIFAR100, in addition to a completely black image sample. bpd is calculated from the density produced by a Glow (Kingma & Dhariwal, 2018) model pretrained on 80 Million Tiny Images, a PixelCNN model trained on CIFAR10, and a diffusion model trained on CIFAR10.

that the combination of the $\gamma$-scaled activation function and a Scaled Weight Standardized layer is variance preserving." Finally, the output of the residual branch is downscaled by 0.2, so the function to compute the output becomes $h_{i+1} = h_i + 0.2 \cdot f_i(h_i)$, where $h_i$ denotes the inputs to the $i^{th}$ residual block, and $f_i$ denotes the function computed by the $i^{th}$ residual branch. Unlike Brock et al. (2021), we did not multiply scalars $\beta_i$ with the input of the residual branch or learned zero-initialized scalars to multiply with the output of the residual branch, as we did not find these two parts helpful in our setting. We also did not attempt to use Stochastic Depth (Huang et al., 2016), which may further improve upon the accuracies reported here. Due to our small batch sizes (32), we also did not use adaptive gradient clipping.

With this setup, the normalizer-free Wide ResNet with depth = 28 and widen_factor = 10 reached 94.0% on CIFAR10. To enable faster experiments, we instead use a smaller architecture with depth = 16 and widen_factor = 2 which reached 91.2% on CIFAR10.

### S2.2 Simplifier Network

For the simplifier network, we extend the Wide ResNet architecture with 3 output heads at different stages of the network that output encodings to be inverted by our generative model, a three-scale Glow architecture as mentioned in Section 2. This leverages the learnt decoder of the Glow architecture for our simplifier. Concretely, our simplifier network predicts scaling and translation coefficients for the Glow encodings of the original image. We invert the scaled and translated encodings using the same Glow network to obtain our simplified images.

## S3 PNG-compressed File Sizes of Simplified Images

We validated that simplified images actually occupy less storage space. We saved images in the PNG file format and calculated the file size. Figure S2 shows the average PNG file size of the simplified images obtained from our simplification framework. Concretely, we PNG-compressed the images at the end of the joint simplification and classification training and then computed the average file size. Varying the upper

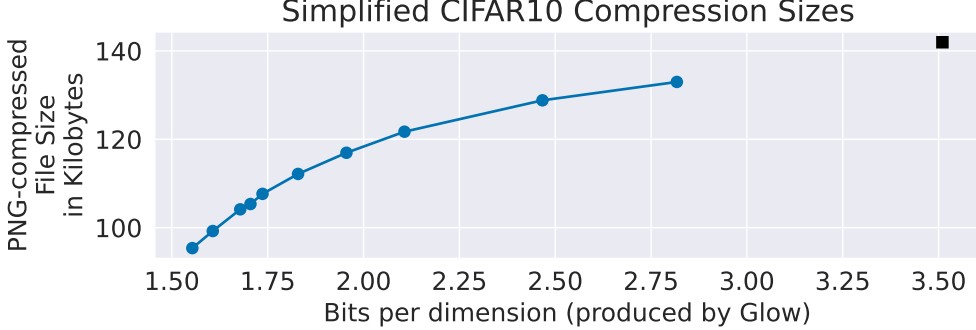

Figure S2: Simplified CIFAR10 images result in smaller storage space when converted into PNG files. Plotted are the average PNG file sizes of the simplified images after joint simplification and classification training (see Section 4), against bpd values produced by Glow, the generative model. Runs with different simplification loss weight $\lambda_{\mathrm{sim}}$ lead to different average file sizes.

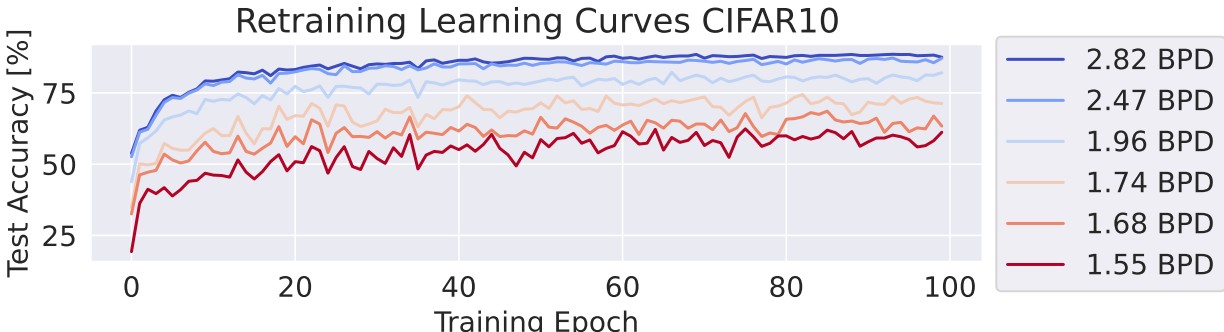

Figure S3: Learning curves for retraining on simplified images on CIFAR10.

bound on $L_{\mathrm{task}}$ leads to different average bpd values and also different file sizes, with larger bounds resulting in smaller file sizes.

## S4 Learning Curves for Retraining

Figure S3 shows learning curves during retraining on the simplified images on CIFAR10. There are no noticeable differences in training speed for more or less simplified images.

## S5 Simplifier Baselines

We implemented three simpler baselines to check whether the losses used in *SimpleBits* during training help retain task-relevant information. In the first baseline, we train the simplifier to simultaneously reduce bpd of the simplified image and the mean squared error between the simplified and the original image. Afterwards we train the classifier on the simplified images and evaluate on the original images the same way as during retraining of *SimpleBits*. In the second baseline, we blur the original images with a gaussian kernel, which also reduces their bpd. We vary the sigma/standard deviation for the gaussian kernel to trade off smoothness and task-informativeness. In the third baseline, we use lossy JPEG compression with varying quality levels. As in *SimpleBits* and the other baselines, we estimate the bits per dimension of the lossy-JPEG-compressed images through our pretrained Glow network for a fair comparison. The gaussian blurring and JPEG compression each replace the simplifier, so these are fixed simplifier baselines without

Per-Instance Simplification During Training, Retraining Results

Figure S4: Comparison between *SimpleBits* and two simpler baselines: In the first one, the simplifier network is trained to simultaneously reduce bpd of the simplified image and the mean squared error between the simplified and the original image. In the second one, gaussian blurring is applied to the input images, different runs vary in the standard deviation used to create the gaussian blurring kernel. In the third one, we use JPEG compression with varying quality levels. Tradeoff curves are mostly worse for the baselines than for *SimpleBits* .

training a simplifier. While these three baselines also retain some task-relevant information allowing the classifier to retain above-chance accuracies (see Figure S4), the tradeoff between bpd and accuracy is mostly worse than for *SimpleBits*, especially on CIFAR10 and SVHN. This shows the losses used in *SimpleBits* help retain more task-relevant information compared to these baselines.

## S6   More Images Simplified During Training

We show a larger number of images that were simplified on CIFAR10 during training with the largest simplification loss weight $\lambda_{\text{sim}} = 2.0$ in Figures S5 and S6.

## S7   SimpleBits with Vision Transformers

To validate that *SimpleBits* also works with non-convolutional architectures, we trained a Vision Transformer (ViT) with *SimpleBits* on CIFAR10 [1]. As Figure S7 shows, while accuracies are lower on CIFAR10 as expected for training a ViT from scratch, *SimpleBits* still shows a accuracy - bits per dimension tradeoff and the simplified images qualitatively contain discriminative information. Interestingly, they look similar, yet different from the simplified images resulting when applying *SimpleBits* to convolutional networks. This suggests the simplified images may reveal what information different architectures are more likely to learn.

Additionally, we also evaluated how the bits-per-dimension/accuracy tradeoff changes when we retrain one type of architecture (Vision Transformers or residual convolutional networks) on simplified images obtained from applying *SimpleBits* to the other type of architecture. Figure S8 show that in all variants on CIFAR10, the networks learn above-chance accuracies, indicating that the simplified images always contain discriminative information that both architectures can exploit. When retraining the convolutional networks, there is a larger decrease in accuracy when using the simplified images obtained via vision transformers. This may indicate that the simplified images of vision transformers are more architecture-specific. An interesting future work can be to more deeply investigate what the simplified images reveal about the inductive biases of different types of network architectures.

---

[1] Implementation taken from `https://github.com/omihub777/ViT-CIFAR/blob/f5c8f122b4a825bf284bc9b471ec895cc9f847ae/vit.py`

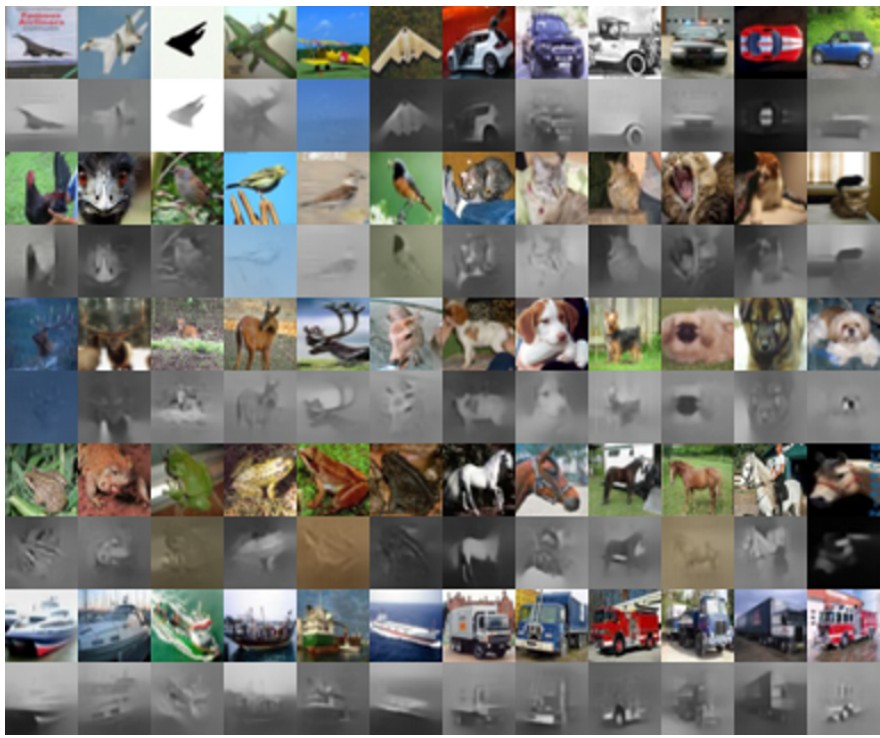

Figure S5: Uncurated set of simplified images with $\lambda_{\text{sim}} = 2.0$, 6 per class. Rows alternate between original and simplified images.

## S8 SimpleBits applied to Dataset Condensation

Now we investigate how *SimpleBits* affects training on a small synthetic condensed dataset. Multiple methods have been developed for dataset condensation (Zhao et al., 2021; Zhao & Bilen, 2021; Wang et al., 2018; Maclaurin et al., 2015), via backpropagation through training (Wang et al., 2018; Maclaurin et al., 2015), gradient matching (Zhao & Bilen, 2021), or kernel based meta-learning (Nguyen et al., 2021). Due to its small size, one can visualize the full condensed dataset to understand what information is preserved for learning. Our aim here is to combine *SimpleBits* with dataset condensation to see if we could obtain a both smaller and simpler training dataset than the original.

In this setting, we jointly condense our training dataset to a smaller number of synthetic training inputs and simplify the synthetic inputs according to our simplification loss (Eq. 1). Concretely, we add the simplification loss $L_{\text{sim}}$ to the gradient matching loss proposed by Zhao & Bilen (2021). The gradient matching loss computes the layerwise cosine distance between the gradient of the classification loss wrt. to the classifier parameters $\theta$ produced by a batch of original images $\boldsymbol{X}_{\text{orig}}$ and a batch of synthetic images $\boldsymbol{X}_{\text{syn}}$:

$$L_{\text{match}}(\boldsymbol{X}_{\text{orig}}, \boldsymbol{X}_{\text{syn}}) = \\ D(\nabla_\theta l(f(\boldsymbol{X}_{\text{orig}}), \boldsymbol{y}), \nabla_\theta l(f(\boldsymbol{X}_{\text{syn}}), \boldsymbol{y})). \tag{S2}$$

where $D$ is the layerwise cosine distance. The matching loss is computed separately per class.

Overall, with our simplification loss, we get:

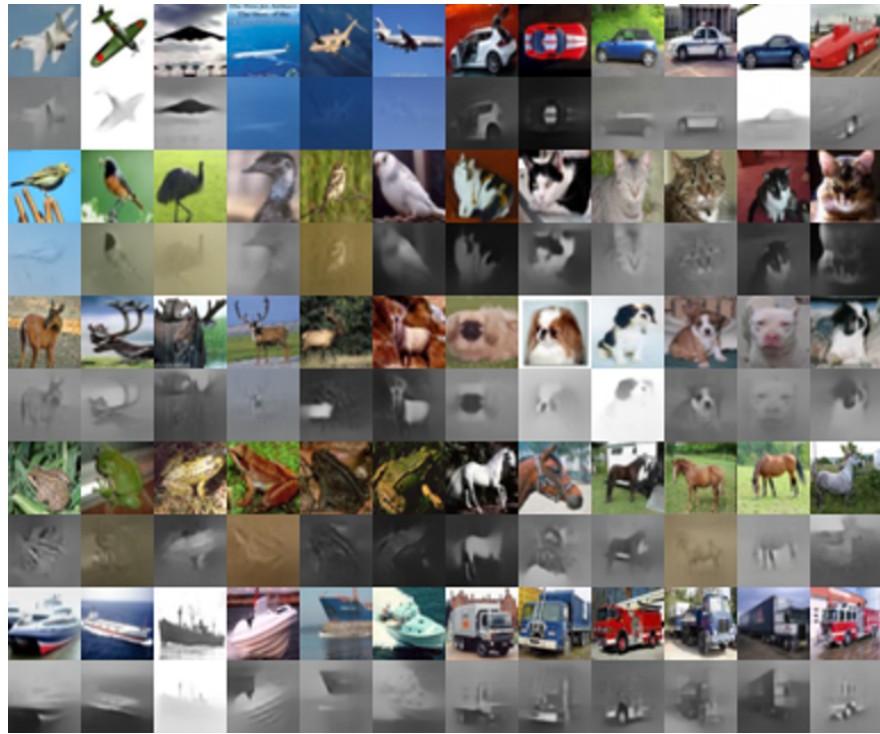

Figure S6: Uncurated set of correctly predicted simplified images with $\lambda_{\mathrm{sim}} = 2.0$, 6 per class. Rows alternate between original and simplified images.

$$L_{\mathrm{syn}}(\boldsymbol{X}_{\mathrm{orig}}, \boldsymbol{X}_{\mathrm{syn}}) = L_{\mathrm{match}}(\boldsymbol{X}_{\mathrm{orig}}, \boldsymbol{X}_{\mathrm{syn}}) \\ + \sum_{\boldsymbol{x}_{\mathrm{syn}} \in \boldsymbol{X}_{\mathrm{syn}}} - \log p_G(\boldsymbol{x}_{\mathrm{syn}}) \tag{S3}$$

We perform dataset condensation on MNIST, Fashion-MNIST, SVHN and CIFAR10 with varying $\lambda_{\mathrm{sim}}$ for the simplification loss. We also apply dataset condensation to the chest radiograph dataset MIMIC-CXR-JPG (Johnson et al., 2019a;b) for predicting pleural effusion and gender. We use the networks from (Zhao et al., 2021), but use Adam (Kingma & Ba, 2015) for optimization.

### S8.1 *SimpleBits* Retains Condensation Performance While Greatly Simplifying Data

In Figure S9, we examine the accuracy for each condensed-and-simplified dataset. We observe that for the natural image datasets, accuracies are mostly retained when decreasing the number of bits per image. Note that the setting with highest bpd is a reimplementation of Zhao et al. (2021) and therefore a baseline without simplification loss. We visualize examples in Figure S9 and observe that the jointly condensed and simplified images look visually smoother, indicating that higher frequency patterns visible in the original images are not needed to reach the same accuracy. These visualizations are also noticeably more smooth than the results for per-instance simplification Figure 5, which suggests that data condensation may already favor features that are less complex. Further condensed sets are in Section S8.2 and a continual learning evaluation in S8.3.

**Evaluation of a Medical Chest Radiograph Dataset** We also evaluate jointly condensing and simplifying for a dataset of chest radiograph images (Johnson et al., 2019a;b). This dataset has known radiologic features for the presence of pleural effusion (Jany & Welte, 2019; Raasch et al., 1982) and difference in gender (Bellemare et al., 2003). In Figure S10, we visualize both the condensed **(top row)** and the jointly condensed and simplified dataset **(bottom row)**. The overlayed shows that a visible difference between

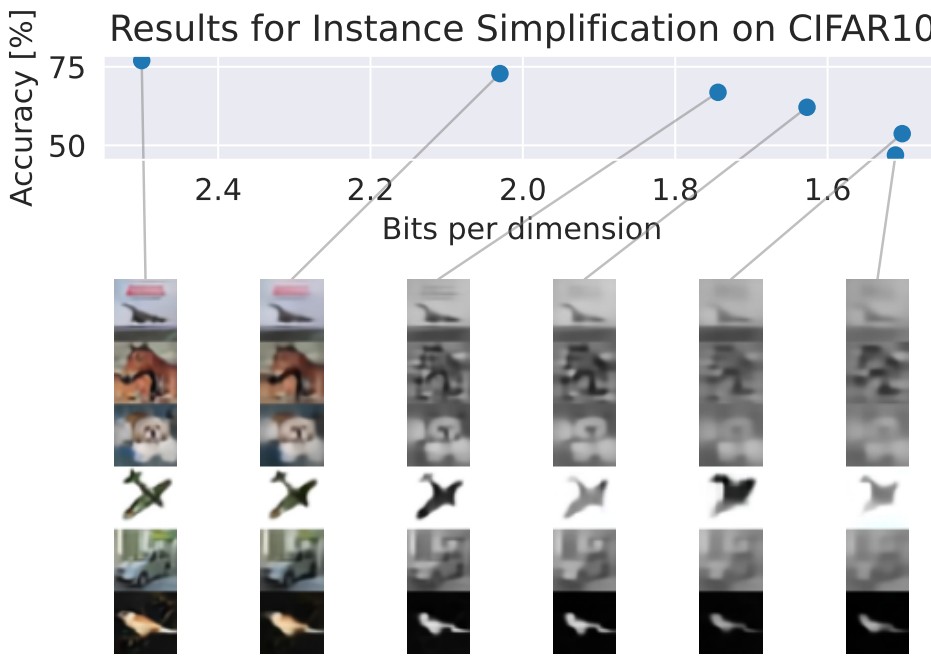

Figure S7: Examples of *SimpleBits* applied to CIFAR10 training images with a Vision Transformer. Conventions as in Figure 2. We observe that simplified images seem to retain discriminative information, yet look slightly different from the ones in Figure 2, suggesting *SimpleBits* SimpleBits reveals architecture-specific discriminative information.

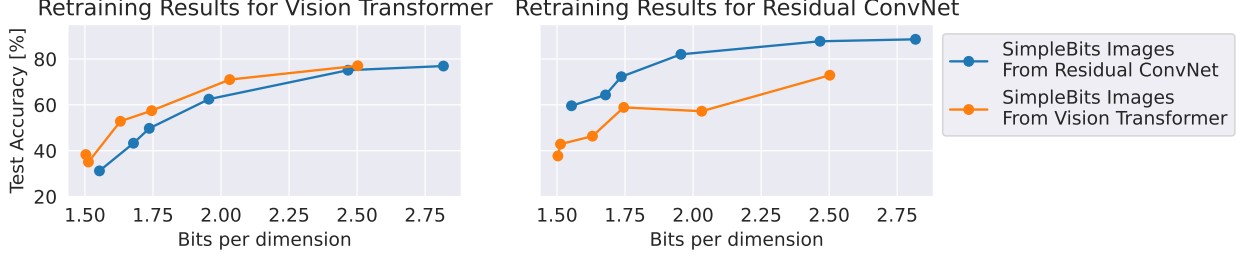

Figure S8: Cross-architecture results retraining an architecture (Vision Transformers or residual convolutional networks) on simplified images resulting SimpleBits applied to another architecture.

presence of feature. For pleural effusion, a larger white region on the bottom of the lung occurs in the simplified pathological image, while for gender, lungs appear slightly smaller for the simplified female image.

### S8.2    More Condensed Datasets

We also show condensed datasets for MNIST and SVHN (Figure S12), some interesting condensed datasets that resulted when we varied the architecture (Figure S13) or the condensation loss (from gradient matching to either negative gradient product or single-training-step unrolling, Figure S14).

### S8.3    Evaluation of Condensed Datasets for Continual Learning

We also evaluate the simplified condensed datasets in a continual learning setting, following (Zhao & Bilen, 2021). In this task-incremental continual learning setting, the model is trained on different classification

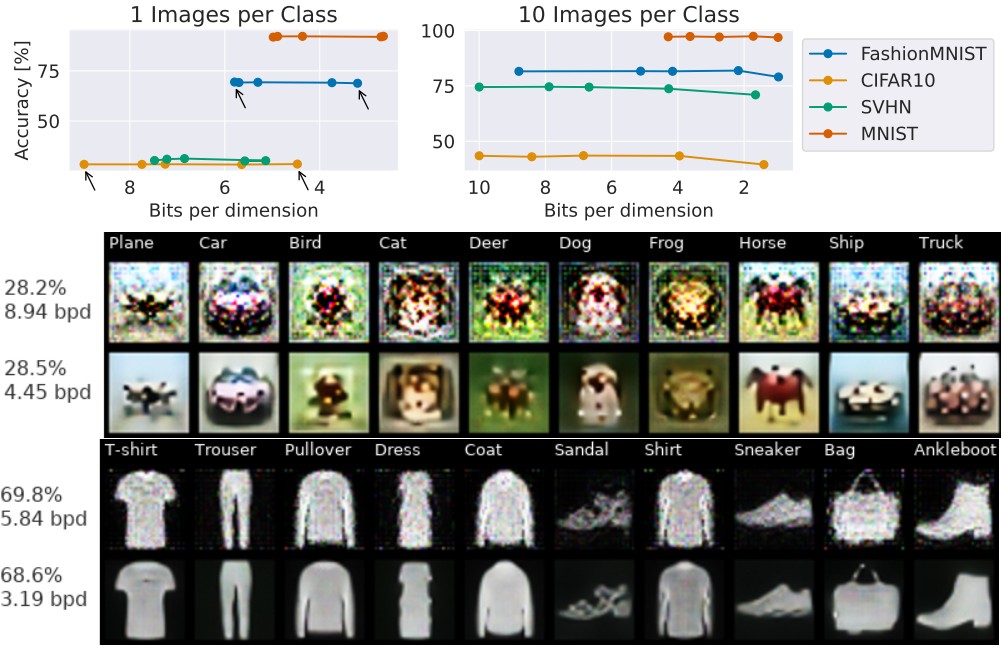

Figure S9: Dataset condensation accuracies (when retraining with the condensed dataset) vs. data simplicity. **Top:** Each dot represents a data condensation experiment run with a particular weight for the simplification loss, which results in more or less complex datasets. Accuracies can be retained even with substantially reduced bits per dimension. In the 1-image-per-class case (**top left figure**), arrows highlight the settings that are visualized in the bottom figure. **Bottom:** Condensed datasets with varying simplification loss weight. Each row represents the whole condensed dataset (1 image per class), with high (top row) or low (bottom row) bits per dimension. Lower bits per dimension datasets are visually simpler and smoother while retaining the accuracy.

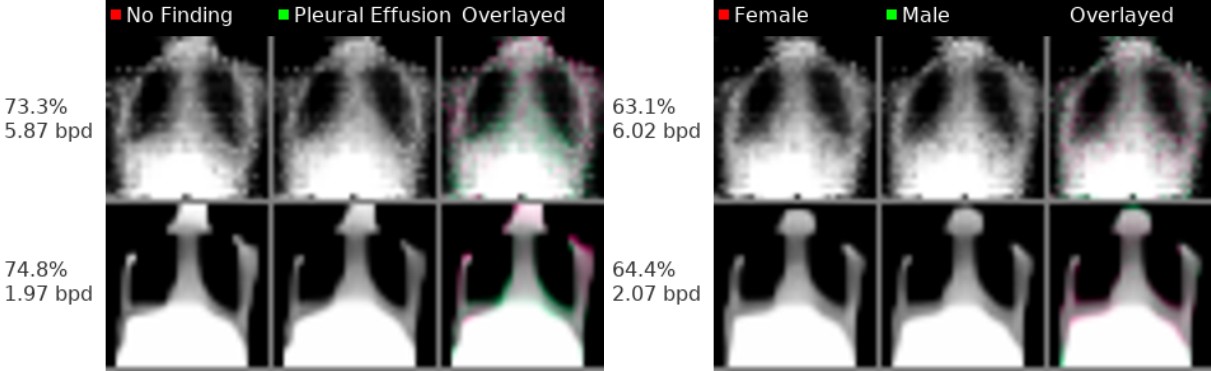

Figure S10: Condensed dataset for pleural effusion and gender prediction from chest radiographs in MIMIC-CXR. Condensed images for the classes look very similar. Color-coded mixed rightmost images reveal the differences between the classes. Green highlighted region at the lower end of the lung consistent with typical radiologic features for pleural effusion (white region indicating fluid on lungs), red highlighted around lung for gender indicate smaller lung volume for the female class.

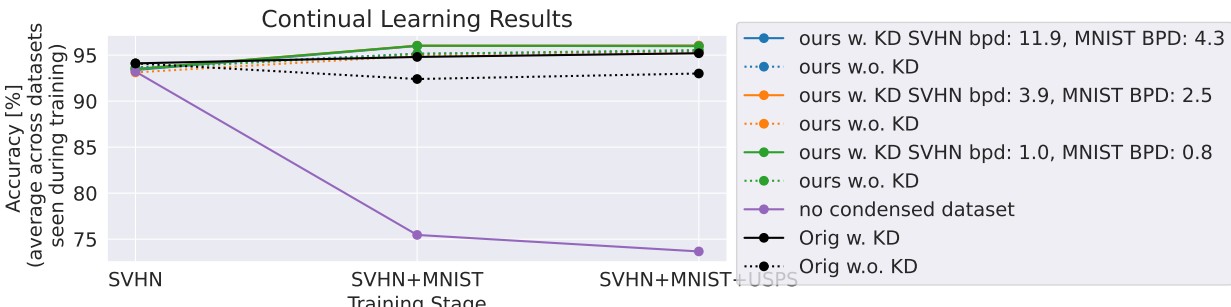

Figure S11: Continual Learning Results without Condensed Dataset (regular sequential training). Conventions as in Figure S15. Accuracies substantially worse without any condensed dataset.

datasets sequentially. When training on a new dataset, the model is additionally trained on the condensed versions of the previous datasets.

The continual learning experiment reproduces the setting from (Zhao & Bilen, 2021) to first train on SVHN, then on MNIST and finally on USPS (Hull, 1994), using the average accuracy across all three datasets of the classifier at the end of training as the final accuracy (see (Zhao & Bilen, 2021) for details).

We created a simpler and faster continual training pipeline that achieves comparable results to Zhao & Bilen (2021). First, we train 3 times for 50 epochs on SVHN, with a cosine annealing learning rate schedule (Loshchilov & Hutter, 2017) that is restarted at each time with $lr = 0.1$. Then for each MNIST and USPS, we train one cosine annealing cycle of 50 epochs for $lr = 0.1$.

We first verified that we can reproduce the prior continual learning results with our simpler training pipeline and find that our training pipeline indeed even slightly outperforms the reported final results (96.0% vs. 95.2% with, and 95.4% vs 93.0% without knowledge distillation) despite slightly inferior performance in the first training stage (before any continual learning, 93.6% vs 94.1%), see following subsection. When using different SVHN and MNIST condensed datasets, we find that we can retain the original continual learning accuracies even with condensed datasets with substantially less (~9x less) bits per dimension (see Fig.S15).

**Without Condensed Dataset** Our training pipeline still exhibits forgetting when not using any condensed datasets of previously trained-on datasets. As Figure S11 shows, the accuracies are far lower than with just regular sequential training. We performed this ablation to ensure forgetting still occurs in our training pipeline.

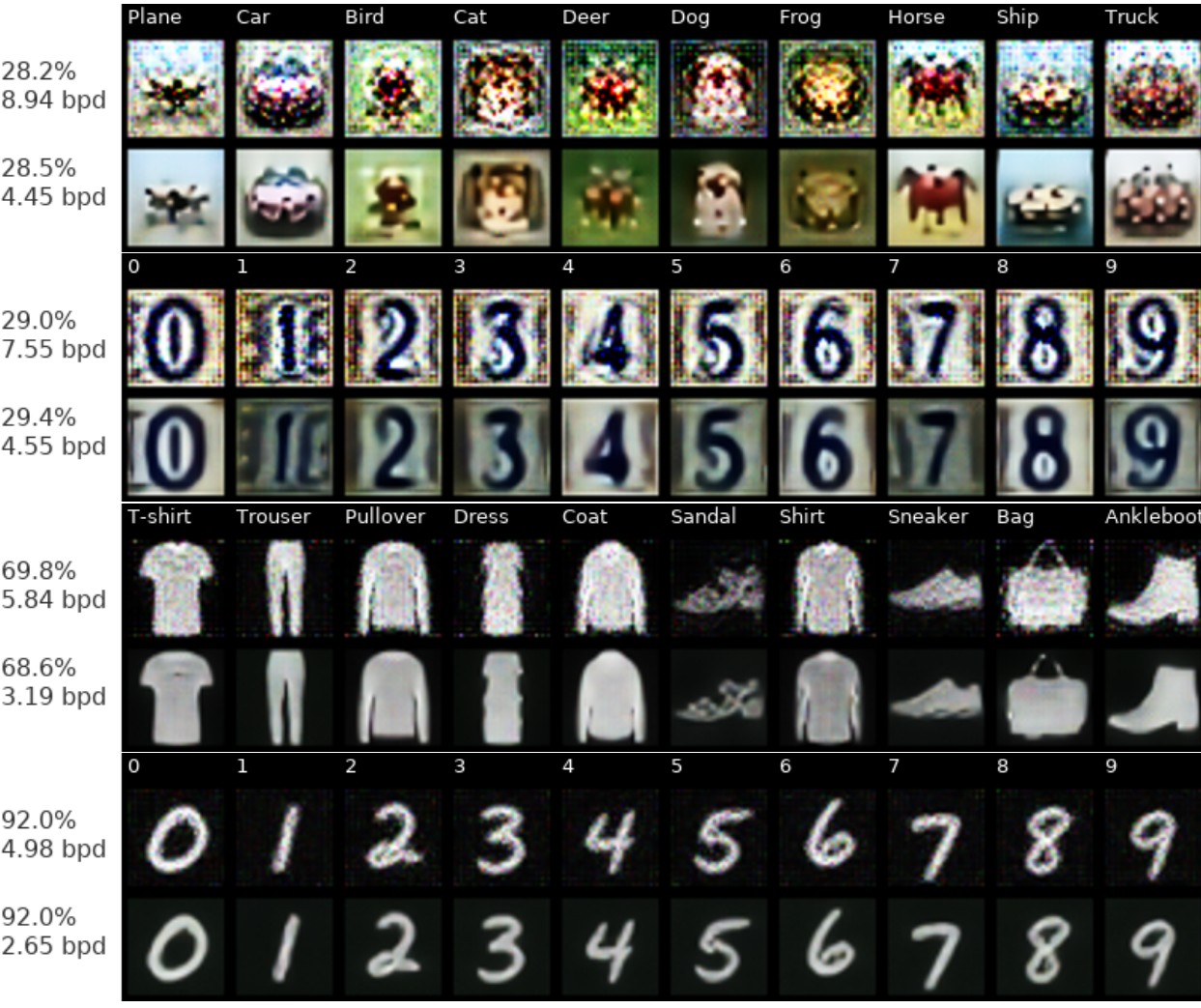

Figure S12: Dataset condensation results with varying simplification loss weight. **Top:** Individual dots represent accuracies for setting with different simplification loss weights. Accuracies can be retained even with substantially reduced bits per dimension. For 1 image per class, arrows highlight the settings that are visualized below. **Below:** Condensed datasets with varying simplification loss weight. Per dataset, showing condensed datasets with high (top row) and low (bottom row) bits per dimension. Lower bits per dimension datasets are visually simpler and smoother while mostly retaining accuracies.

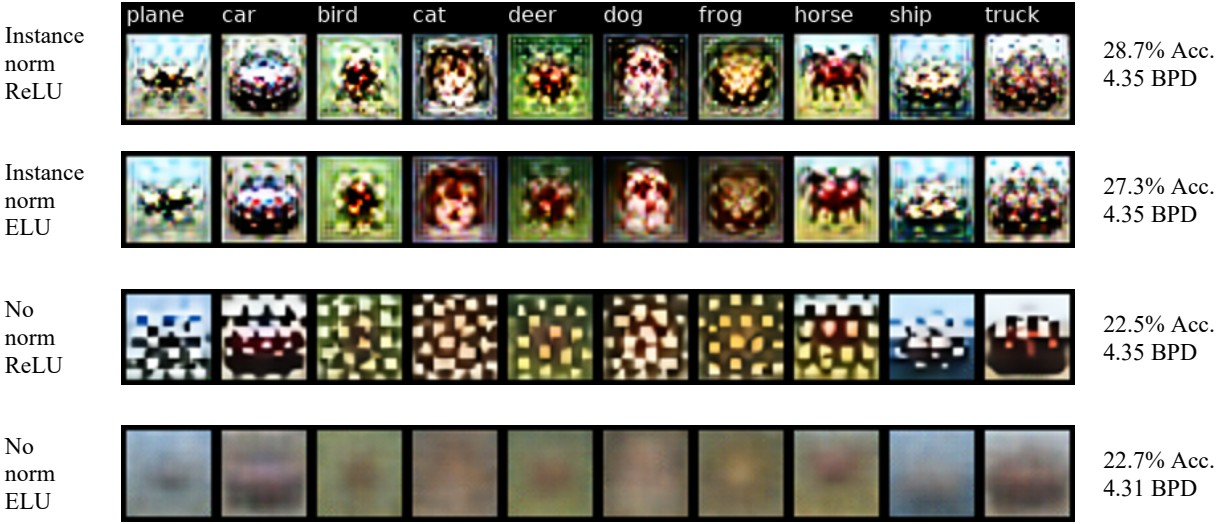

Figure S13: Dataset condensation on CIFAR10 with varying architecture.

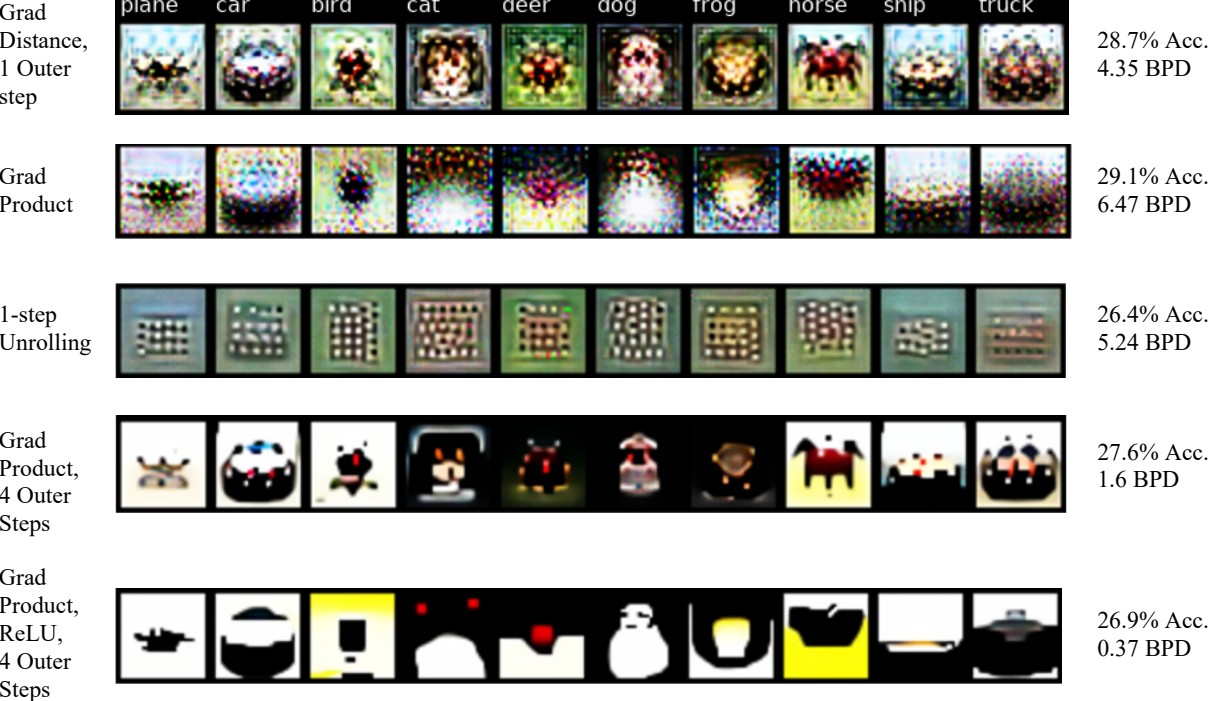

Figure S14: Dataset condensation on CIFAR10 with varying condensation loss and varying outer loop steps, i.e. how many steps the classifier is trained at each training epoch (default 1 in the 1 image per class setting), after each step the condensation loss is again optimized.

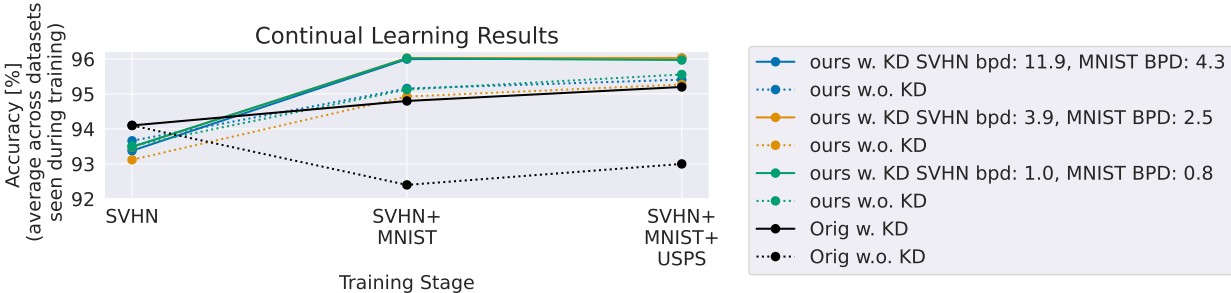

Figure S15: Continual Learning Results. Results for first training on SVHN, then MNIST and then USPS for condensed datasets with varying bits per dimension. Solid lines are with and dashed lines without knowledge distillation. Note that continual learning accuracies remain similar also for substantially reduced bits per dimension. Ablations show that accuracies degrade without any condensed dataset, see supplementary.

