# OpenReview forum: "When Less is More: Simplifying Inputs Aids Neural Network Understanding"
_TMLR — Rejected by TMLR_

### Review · Reviewer_rpVL · 2023-07-28

**Summary Of Contributions:**

The paper investigates how neural network image classifiers respond to simpler inputs and explores their interaction with the learning process. It proposes a framework using input simplicity measures and class-discriminative information optimization during training. Experiments show that simplifying data removes unnecessary information and retains crucial features, enhancing classifier performance. The study also evaluates the impact of simplification on real-world data, showcasing potential practical applications for improving model robustness.

**Audience:**

Yes

**Broader Impact Concerns:**

Not applicable.

**Claims And Evidence:**

Yes

**Requested Changes:**

- Are there any other scenarios that the proposed SimpleBits algorithm can be applied to?

**Strengths And Weaknesses:**

Strengths:
- Overall, the paper is well-written and easy to follow.
- The paper introduces a novel framework to quantify the amount of information present in the data.
- The paper conducts comprehensive experiments to demonstrate the effectiveness of the approach.
- The demonstrated ability of the proposed algorithm to remove information irrelevant to a given task is interesting.

Weaknesses:
- My main complaint regarding the proposed algorithm is its general usefulness. While the authors of the paper demonstrated in Figure 9 that the proposed algorithm is more robust towards some forms of input noises, the improvement is not consistent at all, and the authors of the paper did not benchmark against other forms of noise-robust algorithms. Other than these robustness benefits, there doesn't seem to be any utility associated with the method from a practical standpoint.

---

> ### Author Response · Authors · 2023-08-22
>
> Thanks for the comments and remarking our work introduces a novel framework with comprehensive experiments in a well-written, easy to follow manuscript.
>
> We view our work primarily as an exploration of the concept of simplifying the training data by reducing information as measured by a pretrained generative model. The practical utility of the concept may be in increasing robustness, interpretability or decreasing dataset storage size. Regarding further use cases, we also show in supplementary section S8 that SimpleBits can reduce information in a dataset condensation setting, in practice allowing to further decrease needed storage, we added a sentence in the manuscript referring to that supplementary section. For interpretability, we also see potential of SimpleBits for revealing image features on datasets where the features needed to solve the classification task may be unclear or surprising, in the way we show in section 5.1.
>
> Still, we agree the results with regard to practical utility are somewhat mixed. We nevertheless believe the current results to be worth sharing as they already show interesting properties of what happens if you try to reduce information in the training data the way we do such as the ability to reveal important features and smaller decreases in accuracy in robust evaluation than in original evaluation (or, as discussed, even accuracy increases in robust evaluation). These results can in our view also serve as a good point for further exploration in future research.

---

### Review · Reviewer_SPaq · 2023-08-05

**Summary Of Contributions:**

The paper introduces "SimpleBits," a method that synthesizes simplified inputs for neural network image classifiers. It measures input simplicity using encoding bit size and minimizes it during training while retaining class-discriminative information. The approach naturally removes irrelevant information and retains discriminative features. The trade-off between task performance and input simplification varies with dataset complexity. Overall, SimpleBits aims at offering insights into the learning process.

**Audience:**

Yes

**Claims And Evidence:**

No

**Requested Changes:**

1. I would suggest to try to smooth the text of section 3 and 4.
2. Clarify the balance between the two loss functions.
3. If no quantitative analysis is provided, I would request to tone down the claims, ie. state clearly that these claims come from a qualitative analysis and it is unclear how prevalent they are. Given this concern, I indicated below that the claims made in the paper are not accurate. I would be happy to change after this point is addressed.


**Strengths And Weaknesses:**

I enjoyed very much reading and reviewing this paper.

+The paper explores the interesting topic of discriminative features.

+SimpleBits seems to be effective to extract the discriminative features. The method is intuitive and it is exiting that it leads to simplified images that allow recognition.


In terms of weaknesses I would like to provide the following suggestions:

- Section 3: I found it a bit challenging to follow. The use of colors with the math equations made it a bit confusing and disrupted the reading flow. It would be great if the text and equations could be presented in a more cohesive manner to help readers better understand the concepts without having to jump around.

- Section 4: I noticed that some crucial details are placed in the appendix, which makes it challenging to grasp a clear picture of how the optimization of the task loss and the sim loss would work in practice. It would be beneficial to include those details in Section 4 itself for better comprehension.

- There's no mention of a term to balance the importance of the two losses. Considering different weights for the losses would also add more insight into the model's behavior.

- Qualitative Analysis: While the insights gained from SimpleBits are valuable, it is essential to support them with a statistical analysis to reinforce the findings. Currently, the takeaways seem to rely solely on qualitative inspection of the simplified images. Incorporating some statistical analysis would add more robustness to the conclusions and further validate the approach.

---

> ### Author Response · Authors · 2023-08-22
>
> Thank you for your comments and the remarks that you find SimpleBits intuitive and exciting.
>
> We rewrote section 3 to first more clearly explain our measure and afterwards provide the 2D toy example as a motivation. Do you find it easier to read now? Also, we moved the pseudocode from supplementary to section 4  to make it easier to understand, do you find it more comprehensive now?
>
> Regarding the balance of the importance of the two losses, this is achieved by varying the upper bound of L_task, we tried to now make this more clear in our writing:
> “To achieve a tradeoff between simplification and preservation of task-relevant information, we set an upper bound on the allowed $L_{\mathrm{task}}$. This intuitively defines how much task-relevant information may be lost due to the simplification on each example and we vary it to investigate different tradeoffs.” Note also that we found “This was empirically more stable than using a linear combination of the losses”. This may be related to the problems with using simple weighted combinations of losses as discussed in https://www.engraved.blog/why-machine-learning-algorithms-are-hard-to-tune/
>
> We agree some claims are mostly supported by qualitative analysis. Thanks for alerting us to this, we have toned down the claims in abstract introduction and results and added that they come from qualitative analysis,e.g. in abstract: “For real-world datasets, qualitative analysis suggests the simplified images retain visually discriminative features” Regarding the quantitative analysis, the accuracies after retraining may provide a more objective metric of how much discriminative information has been preserved, although only on the level of the whole dataset, not on the level of individual images.

---

> > ### Comment · Reviewer_SPaq · 2023-09-08
> >
> > Thank you, all is clear now. All my concerns have been addressed and I suggest accepting the manucript.

---

> > > ### Author Response · Authors · 2023-09-09
> > >
> > > Thanks for your positive response! Can you also adjust your response to "Claims and Evidence" appropriately, if you are satisfied in that respect now? :)

---

### Review · Reviewer_tFx6 · 2023-08-20

**Summary Of Contributions:**

- A new method for sample simplicity reduction based on a pretrained generative model and discriminative information from gradients and activations.
- The interesting investigation of the trade-off between input simplicity and task performance.

**Audience:**

Yes

**Claims And Evidence:**

Yes

**Requested Changes:**

Please answer the questions raised in Weaknesses with empirical or theoretical evidence.

**Strengths And Weaknesses:**

Strengths
- The paper forms an intuitive solution to the sample simplicity reduction problem.
- The empirical studies are insightful and convincing.

Weaknesses
- As noted, the task-relevant information is estimated based on an NN, which yields gradients and activations. So, the estimated task-relevant information is dependent on the NN used. I want to know if the estimation keeps consistent across various model selections. Can the sample simplicity estimated given CNNs apply to architectures such as ViTs or MLPs?
- You use GLOW for density estimation. However, it can hardly proceed with high-resolution images due to the inefficiency issue. Have you tried more advanced density estimators like Diffusion Models (DMs)? What is the challenge of doing so?
- It seems that the method cannot handle the unsupervised learning setting. If doing so, the novelty of this paper can be further enhanced. Please clarify this.

---

> ### Author Response · Authors · 2023-08-22
>
> Thanks for your comments and the remarks you find SImpleBits intuitive and that our work contains insightful and convincing empirical studies.
>
> Regarding other classifier architectures, we will try to perform further experiments with other architectures in the rebuttal time.
>
> Regarding efficiency of Glow and potential to use other generative models, indeed figure S1 in the supplementary suggests that bits per dimensions provided by other generative models may also serve as good proxy metrics of simplicity of an image. One route towards higher-resolution images with Glow can be to reduce the number of parameters (e.g., reduce the number of filters and/or layers), as we only need a generative model that can provide a good proxy metric for simplicity of an image, which may not need a very high-performing generative model. For other types of models, Autoregressive models may be an alternative, I presume diffusion models may be a bit slow in terms of computing the bits per dimension.
>
> Indeed, SimpleBits always requires a second information-preservation loss. Without it, any image may be simply mapped to a low-information image like a completely black image. For the unsupervised case, one may try to find another kind of information to preserve, e.g. to preserve the information contained in CLIP (https://arxiv.org/abs/2103.00020) embeddings. So concretely for the unsupervised case, one may try to minimize the distance between the CLIP embedding of an original image and the corresponding simplified image while trying to reduce the BPD of the simplified image. Other schemes may also be envisionable if one can define other types of information to preserve in the unsupervised setting.

---

> > ### Author Response · Authors · 2023-08-31
> >
> > We also now briefly experimented with Vision Transformers and found the following, see updated supplementary (section S7):
> >
> > "To validate that SimpleBits also works with non-convolutional architectures, we trained a Vision Transformer (ViT) with SimpleBits on CIFAR10. As Figure S7 shows, while accuracies are lower on CIFAR10 as expected for training a ViT from scratch, SimpleBits still shows a  accuracy - bits per dimension tradeoff and the simplified images qualitatively contain discriminative information. Interestingly, they look similar, yet different from the simplified images resulting when applying SimpleBits to convolutional networks. This suggests the simplified images may reveal what information different architectures are more likely to learn."

---

### Author Response · Authors · 2023-09-02
**Thanks for all reviews, clarified text and performed additional experiments**

Final general statement as review interaction period is ending:

We thank all reviewers for their thoughtful comments and are delighted they found that our manuscript presents an "intuitive solution to the sample simplicity reduction problem" with "insightful and convincing" empirical studies (Reviewer tFx6), was enjoyable to read and review as an interesting exploration of the topic of discriminative features (Reviewer SPaq) and demonstrated an interesting ability to remove irrelevant information (Reviewer rpVL).

As written in the individual answers, in the rebuttal time, we have rewritten sections 3 and 4 and provided more details for additional clarity and clarified our language for some of the claims of the paper. We also highlighted existing dataset condensation work in the supplementary as a further potential practical use case. Further, we performed additional experiments with vision transformers that show SimpleBits also works on other types of models, showing their  information-accuracy tradeoff and still yielding qualitatively discriminative simplfied images.

We sincerely hope that the novelty of this work and its unique approach of understanding the information deep classifier networks use through simplifying training inputs can serve as a valuable research contribution to TMLR. We believe it fits nicely into the scope of TMLR as a venue for technically solid research that strays a bit further from commonly used methodologies in the ML community and hope our manuscript can inspire further research on the idea of of understanding deep networks through reducing information in their training data.

---

### Decision · Action_Editors · 2023-11-07

**Recommendation:** Reject

**Comment:**

Still, I believe that there is substantial value in this approach, and I would therefore like to encourage the authors to **revise and resubmit their manuscript to TMLR**. Producing the more detailed comparison to other methods for understanding the critical aspects of the input and moving to more quantitative comparisons is beyond a “minor revision” that could be accepted directly without another round of review.

**Audience:**

The subject and approach are very relevant to TMLR's audience. This should be further helped by the revisions suggested above and in the reviews, which will enhance the practical usefulness of SimpleBits and relate it to other methods to full address the questions posed by the authors.

**Claims And Evidence:**

This paper presents an intriguing mechanism for exploring how data instances can be simplified while still retaining task-relevant information. The method is straightforward (which is a good thing), and so can be applied easily to numerous algorithms/datasets/etc. This is shown in the extensive experiments, with particularly interesting results showing that the method can eliminate distractor features of the input. The reviewers were overall positive about the paper, but identified a few shortcomings that would lead to improvement.

The paper introduces two main questions in the abstract: “How do neural network image classifiers respond to simpler and simpler inputs?” and “...what do such responses reveal about the characteristics of the data and their interaction with the learning process?”. The first is clearly answered, but the second could be pushed far more in order to fully support the paper’s claims that SimpleBits reveals insight into learning algorithms. The reviewers identify this in their comments on the "practical usage” or “general usefulness”, and the authors even admit in the discussion that “the results with regard to practical utility are somewhat mixed.” (Despite this, I agree that the results are worth sharing, as the authors pointed out.) The utility of SimpleBits is the insight to which it can bring, such as to which parts of the input are task relevant. These aspects are explored in the paper, but really need to be pushed farther to support the claims (even after they were toned down in the revision).

In particular, there is a whole body of related work on saliency analysis that addresses how input features affect the neural net's decision. This paper addresses a similar question from a slightly different view. Methods from saliency analysis, such as class activation maps (CAM), grad-CAM, etc. provide related insight to SimpleBits. What does SimpleBits provide us that is similar or different to these methods? There’s no comparison against these other methods, and that would seem to be the fundamentally scientifically interesting aspect of the approach — to reveal insights into the learning algorithms. The current evaluation does reveal insights, but they aren’t put into context with how these insights are related to those garnered by other existing approaches.

These comparisons should also be quantitative wherever possible, instead of simply qualitative as in the current draft. To do this, the authors should look at related work to determine the best mechanism for quantitative analysis. Two possibilities may be 1.) use a user study to assess the quality of the output or 2.) do a comparison against a dataset with existing human-attention labels (e.g., here is where eye tracking says people looked to classify this image). It is likely that #1 might be a bit extensive, but #2 should be possible. The authors may also consider quantifying the similarities/difference vs saliency analysis methods by having them analyze the same image.

In addition, even with the revisions, there are still some parts of the paper that could use further clarification. In particular, Section 3 could be made more clear (I found the explanation a bit convoluted, with some disconnections between adjacent paragraphs) and critical details that are in the appendix should be moved into the main text. For example of the latter, the qualitative analysis in 5.4 is especially brief and simply states “the simplified images still retained plausible Chest-X-Ray features for pleural effusion” without further explanation. This is clarified significantly more in the appendix, but would be stronger if the critical features as needed to read these x-rays (or what is in medical illustrations, etc.) are shown to be in the simplified images. Adding in the quantitive analysis suggested above would further strengthen that claim.

**Resubmission Of Major Revision:**

The authors may consider submitting a major revision at a later time.